# Gridshells in Recent Research—A Systematic Mapping Study

**Steinar Hillersøy Dyvik** [1,*] **, Bendik Manum** [1] **and Anders Rønnquist** [2]

1    Department of Architecture and Technology, Norwegian University of Science and Technology, NTNU, 7491 Trondheim, Norway; bendik.manum@ntnu.no

2    Department of Structural Engineering, Norwegian University of Science and Technology, NTNU, 7491 Trondheim, Norway; anders.ronnquist@ntnu.no

\*    Correspondence: steinar.dyvik@ntnu.no; Tel.: +47-47859151

**Abstract:** Gridshells are shells where the structural system is some kind of grid of linear members rather than a surface. With today's focus on environmentally friendly solutions, gridshells have gained increased relevance as inherently material-efficient structures. This paper investigates the recent research on gridshells, who performs it and what their contributions are, and will thus provide an overview of the research field of gridshells. This study is performed as a systematic mapping. The articles were categorised by research type, motivation, contribution, gridshell type, material, and scientific field. The study shows that most articles are within structural engineering, whereas contributions from architecture were hard to find. The typical study was theoretical studies performing analyses on a specific load or structural behaviour. Some possible knowledge gaps were also identified, including review articles on loads and behaviour, research on bending active metal gridshells and development of gridshell nodes.

**Keywords:** gridshells; lattice shells; reticulated shells; systematic mapping

## 1. Introduction

With a world in need of environmentally friendly solutions, shell structures are relevant as a material-efficient way to construct buildings. Traditionally, shell structures demanded extensive manual labour for constructing the doubly curved shapes and were therefore popular as concrete or masonry shells in the period 1925–1975 [1], when materials were more expensive relative to labour than today. The recent development in digital design and digital manufacturing tools, enhanced by today's focus on reducing material usage, can again favour shell structures. One particular type of shell, gridshells, are shells where the structural system is some kind of grid of linear members rather than a surface, often consisting of straight members connected at nodes. Gridshell construction can be favourable regarding prefabrication, transportation, and construction. The filigree construction of a gridshell also possesses aesthetic qualities regarding the curved global shape and the underlying construction. The gridshell members represent the *bones* of the structure, weaved in an exciting interplay of one or several materials, revealing the structural logic of the shell [2] (p. xvii), a quality often referred to as tectonics in architecture. The knots or joints connecting the bones of the gridshell could further enhance the material transition and structural sense.

Regarding the design of shells, it is said that the distinction between architect and engineer is almost artificial [3] and unclear who actually owns the design [2]. This study is motivated by an interest in the design of gridshells and investigates recent research in order to overview the status in the field, paying particular attention to the contributions from different professions in research. The study is conducted as a systematic mapping study. Some examples of reviews of gridshell research are focusing on the production and market [4] and the design and construction of kinematic gridshells [5,6], reticulated aluminium shells [7] and form and strength optimisation [8]. A recent review focuses on

the triangle grid as a basis for construction [9]. Compared to the studies mentioned above, this study has a wider scope and maps all types of gridshells, gridshell parts and building materials. The study collects and categorises publications and discusses possible gaps in the research based on this mapping.

### 1.1. Focus and Terms

The following section presents the focus of the article and defines the main terms applied. See also Figure 1. A shell structure is a curved surface structure that derives its strength from its geometry, typically constructed from timber, metal, concrete or masonry. Shells are generally thin in the direction perpendicular to the surface, and work through a combination of membrane (mainly) and bending action [10] (p. 31). A Gridshell, lattice shell or reticulated shell is a general term for a shell constructed from some kind of grid of linear members rather than a surface. The terms are used inconsequently. Gridshell is most often used to describe those built with timber and bent during erection (bending active), while the term reticulated shell is more typically used for metal structures, often barrel vaults, spherical caps, or braced domes, for industrial purposes.

Bending active gridshell, post formed, kinematic, elastic, active-bent or strained is a gridshell where the members are initially flat and bent into shape. The grid is typically quadrilateral, with one or several layers in each direction. After bending, the members are locked from further moving, typically by tightening connections and adding bracing as diagonal members or cables for providing in-plane shear stiffness. The most well-known example is probably the Mannheim Multihalle by Frei Otto [11]. A gridshell is often termed discrete gridshell or rigid gridshell when it consists of straight members connected at a node. As opposed to a bending active gridshell, the nodes typically cater for the geometrical changes between members, horizontal angle, tilt angle and torsion, depending on the global shape and topology. When discussing shape, global shape or just shape describe the overall geometry of the entire shell, representing both the building envelope as an architectural element and the structural system.

Topology, network, or grid pattern are terms for the logic and order of connectivity between the members of a gridshell. The most common topologies are quadrilateral and triangular; however, numerous topologies are possible. The topology contains information about the members and the nodes that connect them, and in a mesh representation, this corresponds to edges and vertices. A member is the strip of material that can be either continuous, like in a bending active gridshell, or shorter pieces between two nodes, like in a discrete gridshell. The node, joint, connection or knot is the physical object connecting two or more members. Whereas several members meet at the same point in a digital model, the node is designed to create the connection away from this point to make it physically possible. Nodes are typically constructed in metal regardless of member material because of the strong forces, especially bending moments, it needs to transfer.

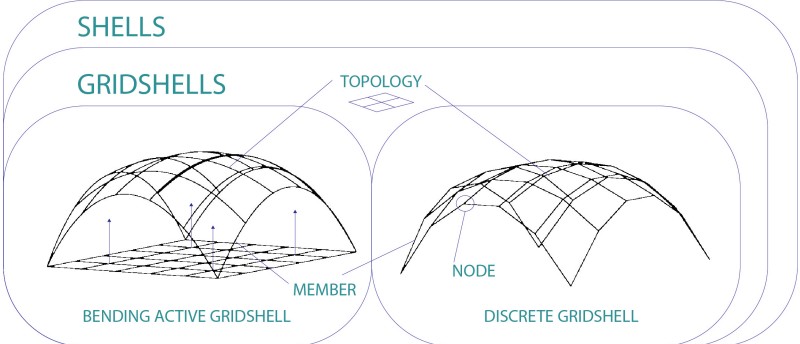

**Figure 1.** Gridshell terminology.

*1.2. Objectives*

This study aims to provide an overview of recent research on gridshells. The following subquestions are defined:

1. What is the amount of peer-reviewed journal papers on gridshells published yearly since 2011, how has this developed over time and what are the most common keywords?
2. What are the scientific fields of the research, and who are the authors?
3. What type of research is conducted?
4. Which types of gridshells, which materials, and which parts of the gridshells are investigated?

The methodological steps will be explained in the following section. The classification results and thus the attempts to answer the objective and subquestions are presented in Section 3. Section 4 provides a discussion on findings. Section 5 concludes the study.

## 2. Methods

This section describes the collection of data. A search query is defined and used to search several online databases. The results from the queries are then screened, as described further in Section 2.1, in order to filter out irrelevant literature. A classification of the literature and the topics covered in the literature is conducted, as described further in Section 2.2. This classification forms the basis of the systematic mapping. The study method is based on Petersen et al. [12]. The method is not so common in structural engineering and even less in architecture, but more used in the field of software engineering, which the method originates from [12]. Recent examples from other fields exist [13–15]. The systematic mapping supports the identification of knowledge gaps in the literature and does so in a scientific and traceable way.

The literature was gathered from Oria/NTNU, Science Direct, SCOPUS, Engineering Village and Web of Science. The search query was the same in all databases, following the setup from Table 1. The columns WHAT and WITH were joined by the Boolean operator AND, while the elements inside each column were joined by Boolean operator OR, forming the following query: "GRIDSHELL OR GRID SHELL OR RETICULATED SHELL OR LATTICE SHELL) AND (ARCHITECTURE OR BUILDING OR CONSTRUCTION OR STRUCTURE)".

**Table 1.** Keywords and limitations used in the search query.

| WHAT | WITH | LIMITED TO |
|---|---|---|
| Gridshell | Architecture | English language |
| Grid shell | Building | Articles |
| Reticulated shell | Construction | Topic |
| Lattice shell | Structure | |

The final search was carried out on 23 September 2021. The search was limited to the topic (title, abstract and keywords), which the databases allowed in various ways, and when possible, limited to *journal articles*, written in *English*. The exact search queries are found in Table 2. Table 2 also presents the number of results obtained from each database, a total of 2372 hits.

**Table 2.** Number of articles obtained from each database using the search matrix and the exact query used. The query was conducted 23 September 2021.

| Database | Search Results | Query |
|---|---|---|
| Oria/NTNU | 169 | (GRIDSHELL OR "GRID SHELL" OR "LATTICE SHELL" OR "RETICULATED SHELL") AND (BUILDING OR ARCHITECTURE OR CONSTRUCTION OR STRUCTURE) |
| Science Direct | 180 | (GRIDSHELL OR "GRID SHELL" OR "LATTICE SHELL" OR "RETICULATED SHELL") AND (BUILDING OR ARCHITECTURE OR CONSTRUCTION OR STRUCTURE) |
| SCOPUS | 845 | TITLE-ABS-KEY ("GRIDSHELL*" OR "GRID SHELL" OR "LATTICE SHELL" OR "RETICULATED SHELL") AND TITLE-ABS-KEY (BUILDING OR ARCHITECTURE OR CONSTRUCTION OR STRUCTURE) AND (LIMIT-TO (LANGUAGE , "ENGLISH")) AND (LIMIT-TO (DOCTYPE , "AR") OR LIMIT-TO ( DOCTYPE , "CP") OR LIMIT-TO (DOCTYPE, "RE")) |
| Engineering Village | 689 | (((GRIDSHELL OR "GRID SHELL" OR "LATTICE SHELL" OR "RETICULATED SHELL") AND (BUILDING OR ARCHITECTURE OR CONSTRUCTION OR STRUCTURE)) WN KY) + ENGLISH3 WN LA + (JA OR CA OR CP) WN DT |
| Web of Science | 489 | TS=((GRIDSHELL OR "GRID SHELL" OR "LATTICE SHELL" OR "RETICULATED SHELL") AND (BUILDING OR ARCHITECTURE OR CONSTRUCTION OR STRUCTURE)) |

### 2.1. Screening

After removing duplicates, a total of 1025 articles were considered for screening. Due to the large number of initial articles (1025), only articles published in peer-reviewed journals were included. This could ensure a better overall quality of the research but may leave out experimental/innovative early-phase research presented at conferences. In accordance with the aim to get an overview of the recent research in the field, only articles published after 2011 were included, leaving 467 articles. Titles and abstracts were read to apply the following inclusion criteria: The title and abstract should include one of the names of gridshell defined in Section 1.1 and describe research on gridshells. Furthermore, the topic of the article should concern gridshells at building scale and consider a shell structure as defined in Section 1.1. The screening was done parallel to the classification. Figure 2 illustrates the steps of the data collection and the screening process. A total of 327 articles passed screening and were used in the classification.

Due to the screening criteria, typically, articles concerning facades, space frames and double layer shells, and hybrid structures, like suspen-dome and prestressed suspended shells, were excluded. Some project presentations were also excluded because the title or abstract did not mention gridshell, even though it is possible that the project was a gridshell, since it was collected with the query.

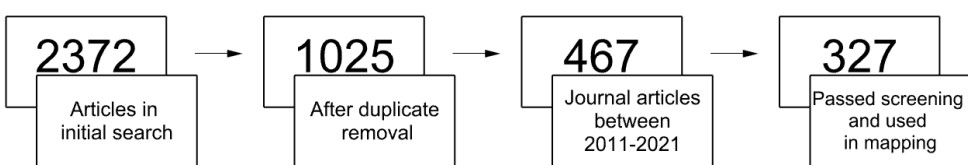

**Figure 2.** Results from the article and data collection process.

### 2.2. Classification

The classification of articles was done simultaneously with the screening. The scheme was, however, adjusted while reading, as illustrated in Figure 3. The classification scheme that was developed for this systematic map is divided into two main parts. Table 3 shows the general classification consisting of research type, research motivation and research contribution. Table 4 shows the second part, which consists of gridshell types, part of the gridshell, and material.

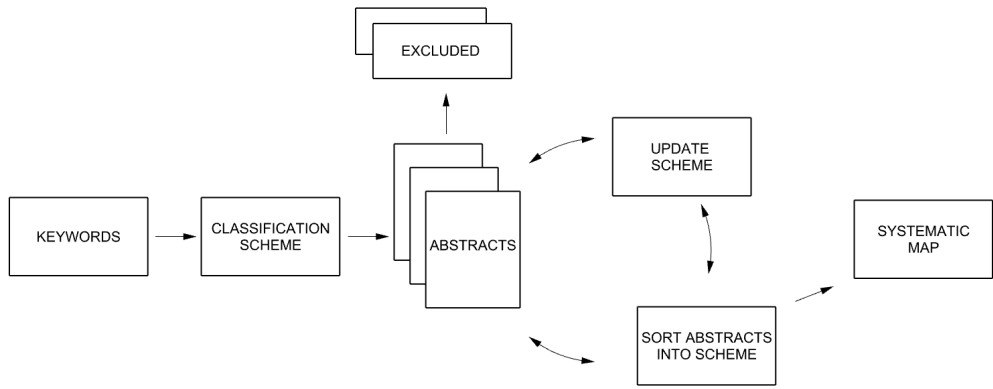

**Figure 3.** Illustration of the systematic mapping process, developed from Petersen et al. [12].

**Table 3.** Classification scheme regarding the scientific field, research type and research contribution for use in the systematic map.

| **Scientific field** | |
| --- | --- |
| Structural engineering | The article concerns the structural design, typically when investigating gridshells at a large scale (global scale or topology) and is published in a structural engineering journal |
| Mechanical engineering | The article concerns mechanical engineering, typically when investigating gridshells at node-scale and is published in a mechanical engineering journal |
| Material science | The article concerns material science typically when developing the material in use |
| Architecture | The article concerns architecture, typically architecture or design is mentioned in the abstract, and the research typically has less focus on the structural performance |
| **Type of research (main)** | |
| Theoretical study | The article presents a theoretical study, typically a numerical analysis |
| Experimental study | The study uses physical testing, typically of a node or a small part of a shell, to evaluate a theoretical analysis |
| Project presentation | The research focuses on the building/structure itself, and the study presents gridshell under construction or built |
| Review article | The study reviews research regarding gridshells or built gridshells |

**Table 3.** *Cont.*

| Research motivation | |
|---|---|
| Loads | The research is motivated by a specific load and how this load affects the gridshell. Examples of this can be loads like seismic-, impact-, temperature-, and explosion loads |
| Behaviour | The research is motivated by a behaviour, like member-buckling or dynamic behaviour, and typically discusses how to improve structural performance related to the behaviour |
| Education of architects/engineers | The research uses the design and construction of gridshells as a case in the education of architects or engineers |
| Design (practice) | The research aims to improve how to build gridshells, typically by improving a method or explaining a process |
| Innovation | The research is motivated by inventing a new product or tool for use in the design or construction of gridshells |
| **Research contribution** | |
| Verification | The study evaluates a theory or estimations by, for instance, physical testing or comparing different methods. |
| Tool | The study proposes a new tool or model. |
| Analyses | The study has performed an analysis and presents the results, typically focusing on a particular load or behaviour. |
| Process | Presents a process like a new construction scheme or procedure for performing analyses. |
| Product | The study presents a physical product, like a new bracing system or node design. |

**Table 4.** Classification scheme regarding gridshell part, material, and type for use in the systematic map.

| Part of gridshell (main) | |
|---|---|
| Global shape | The main focus of the research is typically how to generate, optimise or form-find the global shape or how a particular global shape behaves under a particular load |
| Topology | The main focus of the research is the topology/grid pattern of the gridshell, typically how this affects parts like the geometry of the node, how it affects the behaviour under particular loads, or how the topology can be changed to improve structural performance or simplify node geometry |
| Node | The main focus of the research is the nodes, typically experimental tests of a node or the development of new node designs |
| Member | The main focus of the research is the members, typically buckling behaviour of a single member or material selection for the member |
| All parts | Several parts of the gridshells are investigated, typically in a project presentation describing or considering several or all parts |
| **Gridshell material (members)** | |
| Timber | Typically spruce, oak, glulam (GL32) or bamboo |
| Steel | Typically RHS or CHS |
| Aluminium | Typically extruded aluminium profiles |
| Composite | Typically carbon-reinforced polymers (and not in the meaning of two materials combined) |
| Other | Any other material |
| Not mentioned | Material is not mentioned |

**Table 4.** *Cont.*

| Gridshell type | |
|---|---|
| Bending active | Gridshells where the initially flat members are bent into shape as defined in Section 1.1, like Mannheim Multihalle |
| Smooth | Gridshells constructed from pre-curved members, typically steel or glue-laminated timber like centre Pompidou Metz |
| Discrete | Gridshells where straight members are connected at nodes, defined in Section 1.1, like British Museum Great Court and the Pods Sports Academy |
| Not mentioned | Gridshell type is not mentioned or determined |

## 3. Gridshells in Recent Research

The objective of this study has been to get an overview of the field of gridshell research regarding the scholarly fields (who), the timeline of publications (when), the publication channels (where), and, maybe most important, grasp the content and focus of the research (what). The questions address these objectives, and the results are presented in the following subsections.

*3.1. What Is the Amount of Peer-Reviewed Journal Papers on Gridshells Published Yearly Since 2011, How Has This Developed over Time, and What Are the Most Common Keywords?*

Following the query in the database search and the definitions used in the screening, this study has identified a total of 327 peer-reviewed journal papers concerning gridshells in the last ten years. As seen in Figure 4, the articles from the mapping increased from ten publications in 2011 to 58 in 2020.

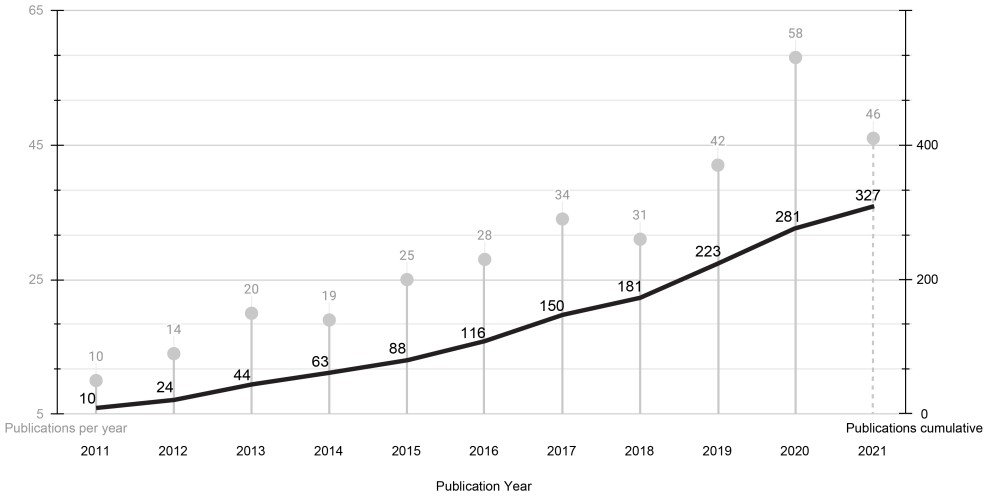

**Figure 4.** Publications per year of the mapped articles. The search was conducted in September 2021 and is therefore not covering the total of 2021.

A brief overview of the content and focus of the gridshell research can be found in Figure 5. Here are the keywords from all articles mapped according to the concurrence, with a minimum of ten co-occurring keywords. This mapping resulted in a total of 54 keywords. The most occurring words can say something about research focus. The five most frequently occurring keywords were shells (structures) (133), structural engineering (44), buckling (42), finite element analysis (32) and structural design (32).

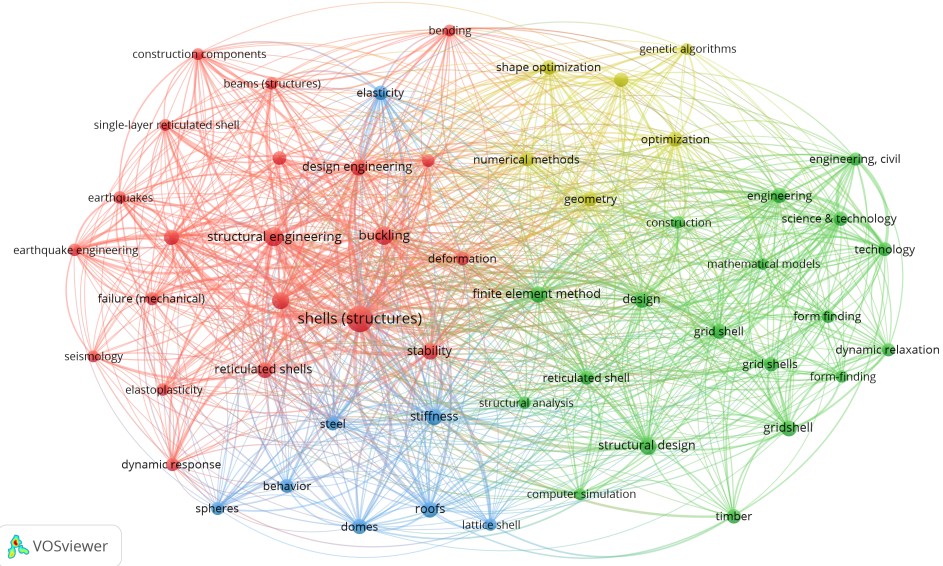

**Figure 5.** Most frequent keywords from all publications mapped according to the concurrence. The colours represent clusters of keywords with higher co-occurrence. The map was created with VOSviewer [16].

### 3.2. What Are the Scientific Fields of the Research and Who Are the Authors?

Figure 6 shows the authors from the systematic map and a mapping of co-authorship. Out of 655 authors, most of them (561) appear in one or two articles. The authors with the most publications were Fan, F. (20), Baverel, O. (20), Liu, H. (18), Zhao, Z. (15) and Douthe, C. (14).

As seen in Figure 7, 278 articles were categorised in the field of structural engineering. The second-largest category was mechanical engineering (25). Few articles were categorised as architectural research (15), and even fewer as material science (8).

Table 5 presents the major publication channels for gridshell research. The top three publication channels were Engineering Structures (Elsevier, h5index 72), Thin-Walled Structures (Elsevier, h5-index 58) and Journal of the International Association for Shell and Spatial Structures (IASS, h5 index 11). As seen in Table 5, the list of publication channels shows many journals based in the structural engineering field, corresponding to the mapping of scholarly fields.

**Table 5.** Publication channels for gridshell research with more than four articles and the number of articles.

| | |
|---|---|
| ENGINEERING STRUCTURES | 31 |
| THIN-WALLED STRUCTURES | 23 |
| JOURNAL OF THE INT. ASSOCIATION FOR SHELL AND SPATIAL STRUCTURES | 22 |
| JOURNAL OF CONSTRUCTIONAL STEEL RESEARCH | 19 |
| INTERNATIONAL JOURNAL OF STEEL STRUCTURES | 13 |
| ADVANCES IN STRUCTURAL ENGINEERING | 10 |
| INTERNATIONAL JOURNAL OF SPACE STRUCTURES | 10 |
| STRUCTURES | 10 |
| JOURNAL OF STRUCTURAL AND CONSTRUCTION ENGINEERING | 9 |
| STAHLBAU | 7 |
| STRUCTURES (OXFORD) | 7 |
| APPLIED MECHANICS AND MATERIALS | 6 |
| AUTOMATION IN CONSTRUCTION | 6 |
| STRUCTURAL AND MULTIDISCIPLINARY OPTIMIZATION | 5 |
| ADVANCED MATERIALS RESEARCH | 4 |
| SHOCK AND VIBRATION | 4 |

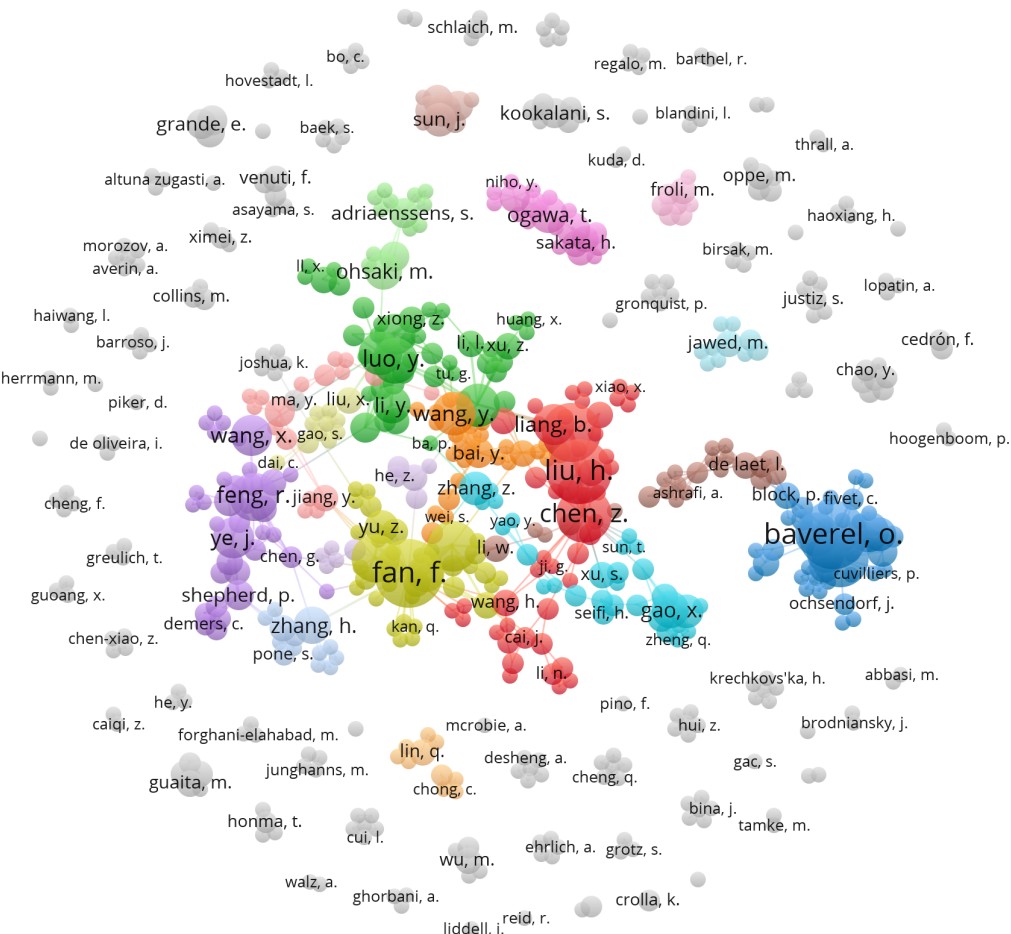

**Figure 6.** Authors of all the publications mapped according to the co-authorship. The colours represent clusters of authors with higher co-authorship. The map was created with VOSviewer [16].

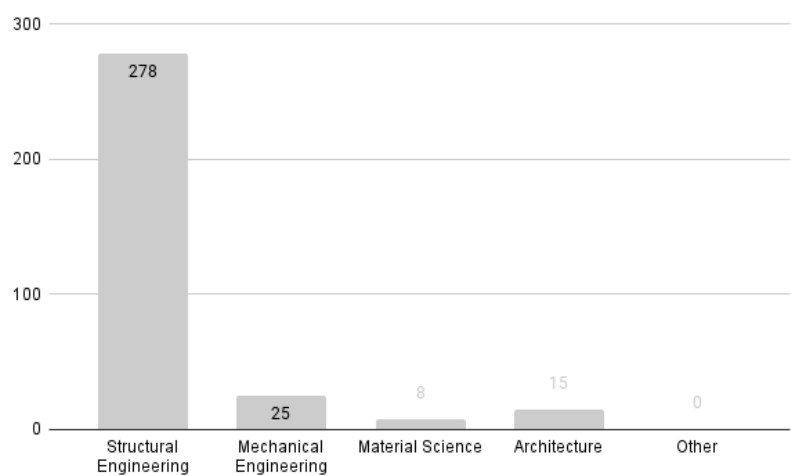

**Figure 7.** Graph showing the scientific field of the mapped articles.

### 3.3. What Type of Research Is Conducted?

The mapping shows that the research on gridshells is mainly structural engineers doing a theoretical study on either behaviour of gridshells or the effect of a particular load or load type. The following section presents findings in the individual facets research contribution, research motivation and research type. Figure 8 shows a combination of all three facets.

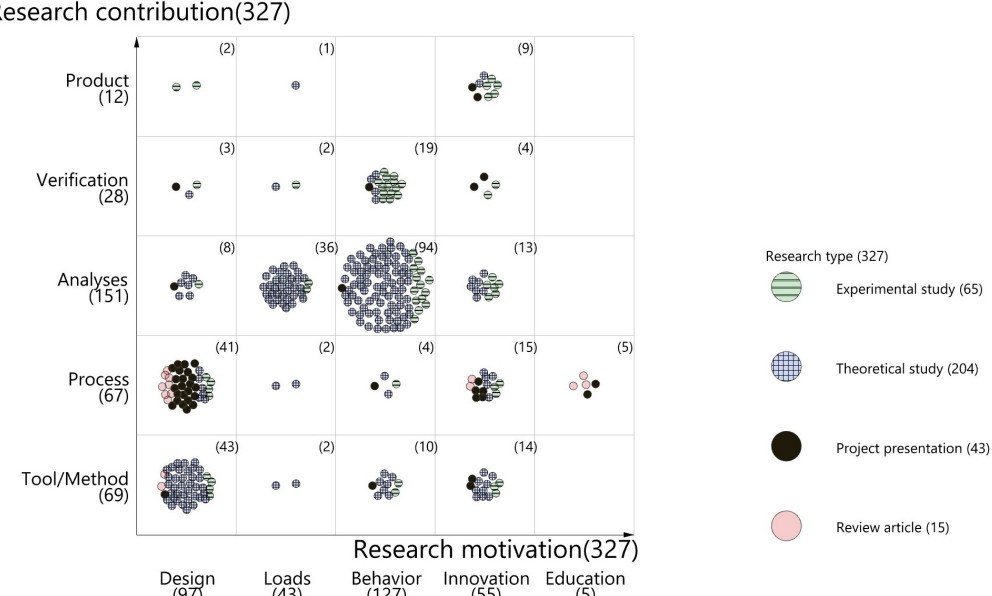

**Figure 8.** Combination of the facets research motivation (*x*-axis), research contribution (*y*-axis) and research type (colour).

### 3.3.1. Research Contributions

The research contribution facet has been mapped in the five categories *verification*, *tool/method* , *analyses*, *process* and *product*, as described in Section 1.1. As seen in Figure 9, the most common contribution was *analyses* with 151 articles, followed by *tool/method* (69), *processes* (67), *verification* (28), and *product* (12).

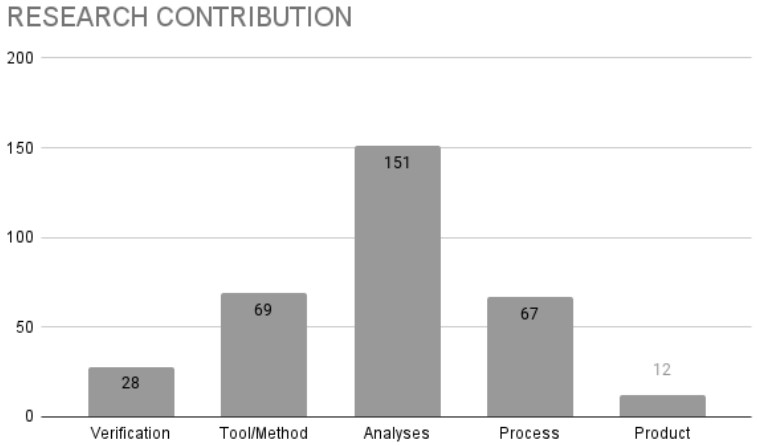

**Figure 9.** Research contribution facet and the number of articles.

As seen in Figure 8, *analyses'* contributions are typically motivated by *behaviour* (94), followed by *loads* (36). Furthermore, the *analyses* are mainly performed as *theoretical studies*. Examples from *analyses* are [17–19]. *Tool/method* contributions are typically motivated by *design practice* (43), followed by *innovation* (14). Furthermore, the *tool/method* contributions are also mainly performed as *theoretical studies*. Examples from *tool/method* are [20–22]. *Process* contributions are also typically motivated by *design practice* (41), followed by *innovation* (15). Furthermore, the *process* contributions are also mainly performed as *project presentations*. Examples from *process* are [23–25]. *Verification* contributions are also typically motivated by *behaviour* (19). Furthermore, the *verification* contributions are also mainly performed as *experimental studies*. Examples from *verification* are [26–28]. *Product* contributions are

almost only motivated by *innovation* (9). Furthermore, the *product* contributions are mainly performed as *experimental studies*. Examples from *products* are [29–31]

### 3.3.2. Research Motivation

The research motivation facet has been mapped in the five categories, *loads*, *behavior*, *education*, *design(practice)* and *innovation*, as described in Section 1.1. As seen in Figure 10, the most common research motivation was *behaviour* with 127 articles, followed by design (97), innovation (55), loads (43) and "education" (5).

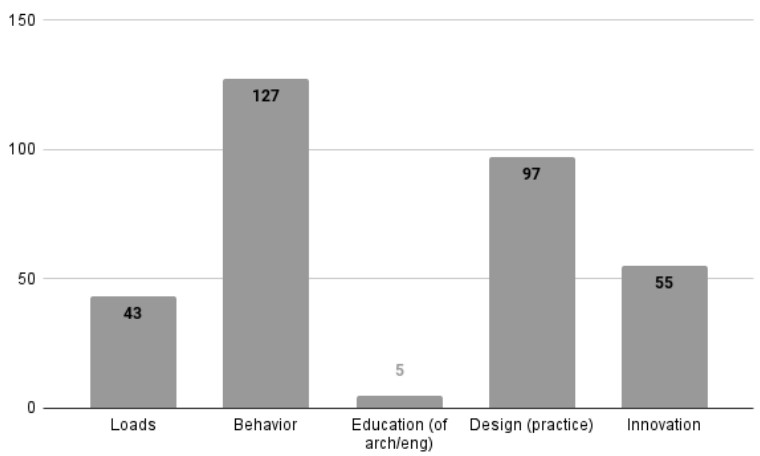

**Figure 10.** Research motivation facet and the number of articles.

As seen in Figure 8, studies motivated by *behaviour* are, as mentioned above, typically contributing with *analyses* (94). Furthermore, the studies motivated by *behaviour* are typically performed as *theoretical studies*. Examples from *behaviour* are [18,26,32]. Studies motivated by *design* typically contribute with *tool/method* (43) or *process* (41). Furthermore, the studies motivated by *design* are typically performed as *theoretical studies* and, to a large extent, *project presentations*. Examples from *design* are [20,23,33]. Studies motivated by *innovation* are quite evenly contributing with either *process* (15), *tool/method* (14) or *analyses* (13). Furthermore, the studies motivated by *innovation* are typically performed as *theoretical studies* or *experimental studies*. Examples from *innovation* are [22,30,34].

Studies motivated by *loads* are typically contributing with *analyses* (36). Furthermore, the studies motivated by *loads* are typically performed as *theoretical studies*. Examples from *loads* are [17,29,35] Studies motivated by *education* are quite a few and only contribute with *process* (5). Furthermore, the studies motivated by *education* are performed as a *review article* or *project presentation*. Examples from *education* are [36–38]

### 3.3.3. Research Type

The research type facet was mapped in four categories, *theoretical study*, *experimental study*, *project presentation*, and *review article*, as described in Section 1.1. As seen in Figure 11, the most common research type found was *theoretical studies* with 204 articles. This was followed by *experimental study* (65), *project presentation* (43), and *review article* (15).

The *theoretical studies*, marked blue in Figure 8, consist mainly of *analyses* or *tool/method* regarding *contributions*. Furthermore, the *theoretical studies* are mainly motivated by *behaviour*. Examples of *theoretical studies* are [17,39,40]. The *experimental studies*, marked green in Figure 8, are divided relatively evenly amongst the contribution facets, with an overweight on the *analyses*. The experimental studies are also mainly motivated by *behaviour*. Examples of *experimental studies* are [19,26,30]. The *project presentations*, marked black in Figure 8, almost exclusively provide contributions regarding the *process*. Furthermore, the project presentations are mainly motivated by the *design practice*. Examples of *project presentations* are 2,8,9. The *review articles*, marked red in Figure 8, mainly provide contribu-

tions regarding the *process* and are mainly motivated by the *design practice*. Examples of *review articles* are [20,24,34].

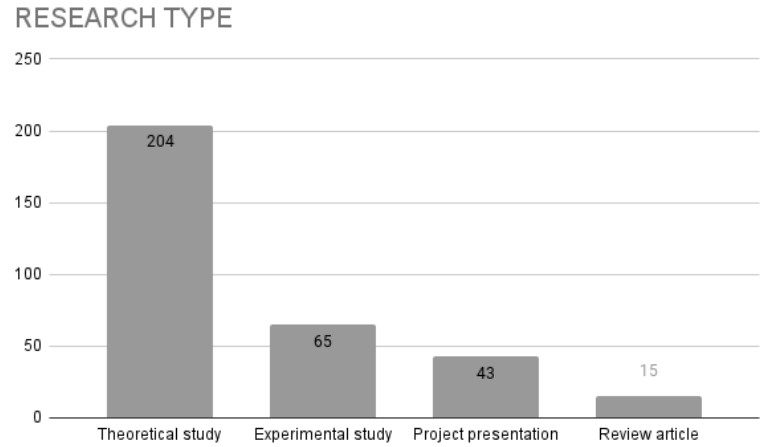

**Figure 11.** Research type facet and the number of articles.

*3.4. Which Types of Gridshells, Which Materials, and Which Parts of the Gridshells Are Investigated?*

The most typical study regarding the gridshell part is on *global shape*, without mentioning material or type, as seen in Figure 12. Moreover, most articles (176) did not mention which type of gridshell it investigated; however, in those studies where the type was decided, it is also typically evident which material and part were studied. The following section presents findings in the individual research facets—materials, parts and type.

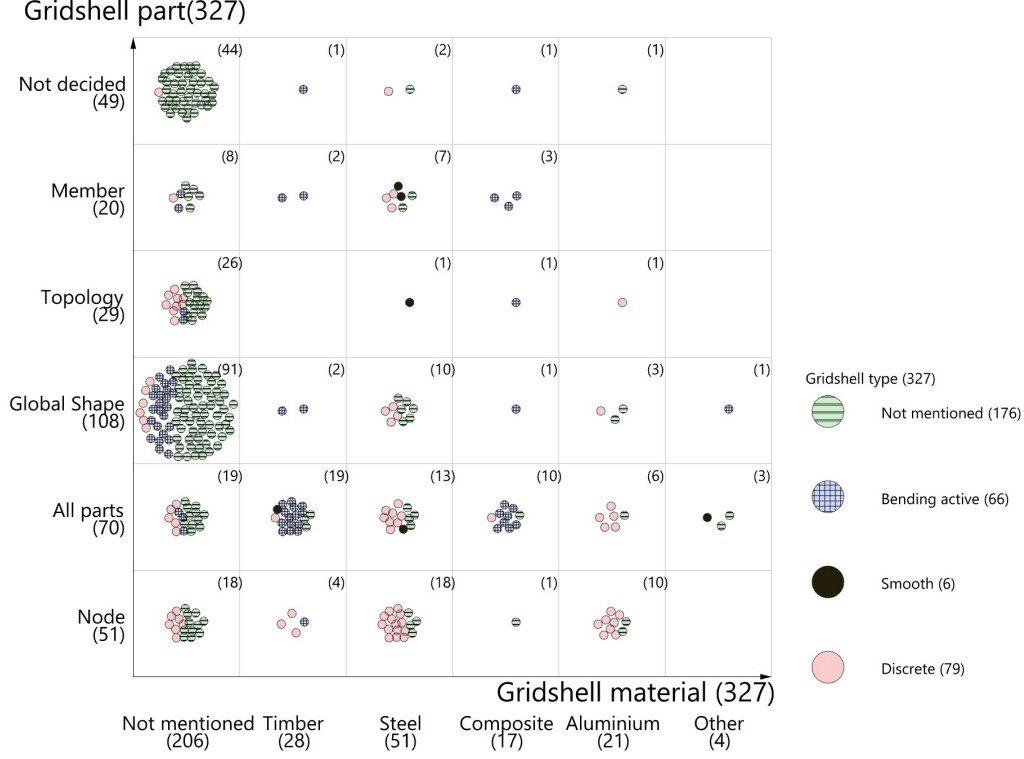

**Figure 12.** Combination of the facets gridshell part (vertical axis), gridshell material (horizontal axis) and gridshell type (colour).

### 3.4.1. Gridshell Material

The gridshell material facet was mapped in the six categories, *timber*, *steel*, *aluminium*, *composite*, *other* and *not mentioned*, as described in Section 1.1. The material facet describes the member material. As seen in Figure 13, the material is typically *not mentioned* (206) in the studies. *Steel* (51) was, therefore, the most used material, followed by *timber* (28), *aluminium* (21), *composite* (17) and *others* (4). Regarding material, classification on the node-material was also done, but this resulted in even fewer articles where the material could be determined, and this facet was therefore excluded. When node material was determined, it was also almost always the same as the member material, with the exception of timber gridshells with metal nodes.

**Figure 13.** Gridshell material facet and the number of articles.

As seen in Figure 12, *steel* gridshells are typically classified as *discrete*. Regarding gridshell parts, studies on *steel* gridshells focus on either the *node* (18), *all parts* (13) or the *global shape* (10). Examples of *steel* studies are [41–44]. *Timber* gridshells are typically classified as *bending active*. Regarding gridshell parts, studies on *timber* gridshells focus on either *all parts* (19) or *nodes* (4)". Examples of *timber* studies are [23,36,45,46]. *Aluminium* gridshells are typically classified as *discrete*. Regarding gridshell parts, studies on *aluminium* gridshells focus typically on the *node* (10). Examples of *aluminium* studies are [30,47,48]. *Composite* gridshells are almost only classified as *bending-active*. Regarding gridshell parts, studies on *composite* gridshells typically focus on *all parts* (10). Examples of *composite* studies are [27,49,50]. *Others* are very few with no clear trends, except that they focus on *all parts* (4). Examples of *other* studies are [51–53]. Examples of studies where the material is not mentioned include [17,18,20]

### 3.4.2. Gridshell Part

The gridshell part facet was mapped in the five categories, *global shape*, *topology*, *node*, *member* and *all parts*, as described in Section 1.1. As seen in Figure 14, the part studied is typically the *global shape* with 108 studies. This is followed by studies on *all parts* (70), *node* (51), *topology* (29) and *member* (20). There were also 49 studies where the part could not be decided.

As seen in Figure 12, studies on *global shapes* (108) were mostly *bending active* regarding gridshell type, although most have the *not mentioned* type. Regarding gridshell materials, studies on *global shape* do normally not specify the material, although some studies are classified as *steel* (10). Examples from *global shape* are [54–56]. Studies on the *node* (51) are mostly *discrete* regarding gridshell type. Regarding gridshell materials, studies on *node* are either on *steel* (12) or *aluminium* (10). Examples from *node* are [20,57,58]. Although most have *not mentioned* the type, studies on *topology* (29) are also mostly *discrete* regarding gridshell type. Regarding gridshell materials, studies on *topology* have typically *not mentioned* (26)

materials either. Examples from *topology* are [59–61]. Studies on *member* (20) are also mostly *bending active* regarding gridshell type. Regarding gridshell materials, studies on the *member* are typically *steel* (7). Examples from *member* are [26,62,63].

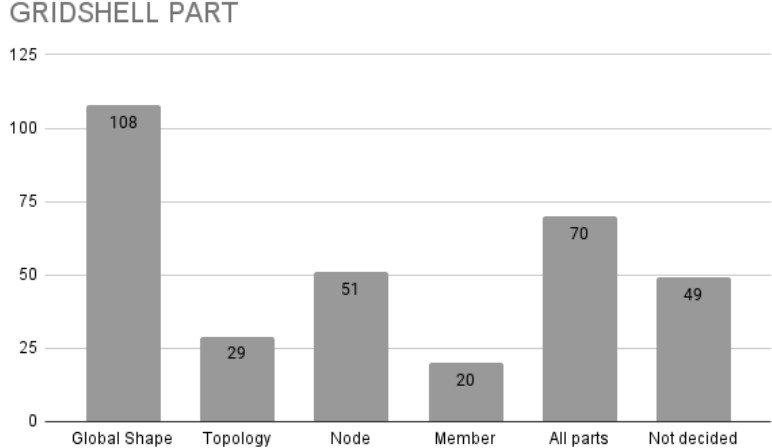

**Figure 14.** Gridshell part facet and the number of articles.

### 3.4.3. Gridshell Type

The gridshell type facet was mapped in the four categories, *bending active*, *smooth*, *discrete* and *not mentioned*, as described in Section 1.1. As seen in Figure 15, the type studied is typically *discrete* with 79 studies. This is followed by studies on *bending active* gridshells (66) and *smooth* (6). There were, however, 176 studies where the type was *not mentioned* or not possible to determine.

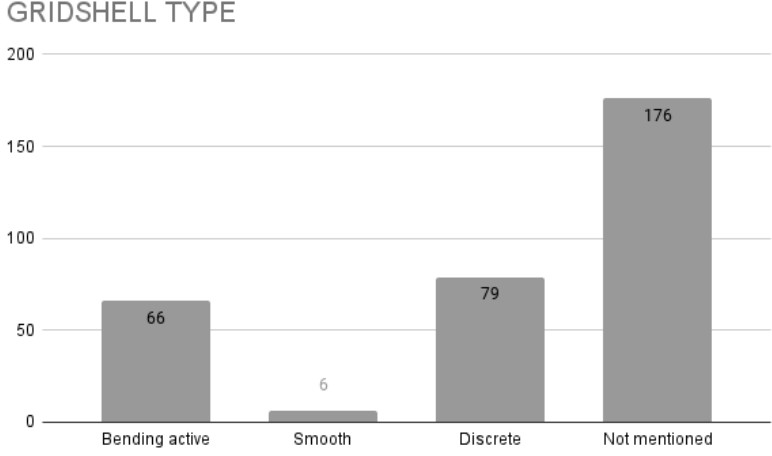

**Figure 15.** Gridshell type facet and the number of articles.

The *bending active* gridshells, marked in blue in Figure 12, typically focus on *all parts* or the *global shape* regarding the gridshell part. Furthermore, the studies on *bending active* gridshells are mostly *timber* or *composite* regarding material. Examples from the *bending active* facet are [21,24,34]. The *discrete* gridshells, marked in red in Figure 12, typically focus on *all parts* or the *node* regarding the gridshell part. Furthermore, the studies on *discrete* gridshells are mostly *steel* regarding material. Examples from the *discrete* facet are [64–66]. The *smooth* gridshells, marked black in Figure 12, are relatively few and placed with no particular trends regarding part or material. Examples from the *smooth* facet are [23,67,68]. Examples of studies where the type was *not mentioned* are [54,69,70].

## 4. Discussion

### 4.1. Limitations to the Study

Some aspects should be considered when interpreting the results of this study. The study is not claiming to be a complete collection of all gridshell research, but it is a comprehensive collection of a rather large and general research topic. Some limitations were made during the collection of the data. The query formulation is a compromise between not excluding too many possible relevant articles and limiting the number of articles. More results could be retrieved by allowing more general search words, like "timber shell" or just "lattice dome". In addition, the selection of databases was limited. Publications not indexed in these databases are not included, which could, for instance, exclude *project presentations* that could be found in architectural magazines. However, in this study, they would have been excluded in the screening because of the limitation to journal articles. The limitation to journal articles has also narrowed down the number of articles and possibly removed some less mature and more innovative research. The screening and classification were based on the title and abstract only. By including the full paper, some articles could be included whilst others were excluded from the screening, and the classification could have ended up with other categories. The fact that the full papers are not included also means that the knowledge gaps discussed in the following sections are, to a lesser extent, based on the content of the articles and instead come from the categories with few contributions. This study could therefore fail to find knowledge gaps in the categories that are well covered.

### 4.2. Publication Trend

According to the mapping, the research on gridshell has had a steady increase in publications from 10 in 2011 to 58 in 2020. The increase can indicate an increased interest in the research topic; however, it can also signify a changing/maturing naming convention. The research on gridshells prior to this study has been conducted since around 2017. The naming conventions used in the query are, therefore, most relevant for the most recent research. Significant research, however, has recently been published on gridshells in book format, including Timber Gridshells from 2016 [2], Bamboo Gridshells from 2015 [71], and Shell Structures for Architecture from 2014 [10]. These publications support the identification of an increased interest in and relevance of the topic.

### 4.3. Scientific Fields

The scientific field was almost exclusively found to be *structural engineering* (278 of 327). This was partly an expected result: Firstly, there are many aspects of gridshell design that are mainly a structural engineering topic. Analyses of gridshells, and especially the global shape, is typically performed by structural engineers. As Section 3.3 showed, there are also many approaches that the structural engineers can have to this research. Secondly, other fields, especially architecture, do not necessarily use the term gridshell (or similar) when describing what essentially is a gridshell roof. When describing a building from an architectural perspective, naming the correct structural type is not essential, and terms like glazed roof, reticulated structure, lattice dome or timber dome have also been used. Terms like these were considered in the initial query but were avoided as they also would generate irrelevant results.

### 4.4. Research Types, Contributions and Motivations

As mentioned before, most research is done by *structural engineers* and performed as *analyses* to examine *behaviour* or *loads*. Many of those studies present a traditional structural analysis that normally would have been part of a structural design report. The results from many of the *analyses* on *behavoir* or *loads* can be too specific because they are linked up to a shape and load combination that only apply in that particular study, and it can therefore be hard to use for others. It can be argued that these studies struggle to provide any contribution to the field.

A combination of facets with no or very few articles found in the systematic map can be used to identify knowledge gaps. Some of these are probably less relevant for novel gridshell research. There were for instance no *theoretical studies* on *education* and no *review articles* providing a *product* as its contribution. On the other hand, it was expected to find more contributions of *products*, like gridshell *nodes* which historically were developed as products by companies like Mero [72]. The mapping showed that little research contributed with a *product* (12). However, it is possible that such development mostly happens in the industry and, therefore, is not published as research. Lastly, the study did not find any *review articles* on *loads* or *behaviour*, which could serve as comprehensive guides for the design and understanding of gridshells for engineers and architects. Judging by the very high number of papers that present a *theoretical study* on *loads* or *behaviour*, it could be very beneficial if this information was reviewed, gathered, and compared. Examples of such reviews exist in the form of a book edited by Kato [73], which reports from the IASS working group 8 on buckling behaviour of metal gridshells and space frames, and a PhD dissertation by Malek [74], which studies the effect of topology and shape on the mechanical behaviour of spherical cap and corrugated vault gridshells. However, these works are not that recent (from 2012 and 2014) and different from the suggested review articles.

### 4.5. Gridshell Types, Parts and Materials

Regarding gridshell types, few examples of *discrete* gridshells in *timber* are found, as seen in Figure 12. However, there are recent examples known to the author. Explanations for this can be exemplified by the article from Harris et al. [75]. This conference paper presents the design and construction of a discrete timber gridshell. Although the terms *gridshell*, *lattice or reticulated shell* and *reticulated structure* have all been used (once) in that article, they are not in the abstract. Only the keyword *timber shells* is used to describe the structure type. Several articles like this might have been missed, either because of the naming or because they were published as conference papers. Similar to the previous section, it is possible to look at combinations of the facets with few or no articles to identify knowledge gaps. Figure 12 shows that examples of *bending active steel* or *aluminium* gridshells are not found in the study. If designed correctly (i.e., by finding a method to add shear stiffness to the cross-sections that need to be very thin to make bending possible), *bending active* metal gridshells could have exciting potential. One example not found in the mapping is Schling et al. [76], which constructs a *bending active* metal gridshell on an asymptotic surface. Only four studies focus on *nodes* for *timber* gridshells [77–79], corresponding with the few studies contributing with the *product* in the previous section. Research on nodes is therefore suggested as another knowledge gap. Only one study [61] looks into the *nodes* in *bending active* gridshells. Those *nodes* are typically solved as simple bolts or plates. They have remained almost unchanged since Mannheim Multihalle and could have a potential for new development, such as metal nodes, containing more details than the typical bolts and plates, or as wood–wood connections neglecting the need for any metal. Lastly, more studies on how to generate and optimise *topology* could be helpful for all gridshell types. Although the material selection should have an impact on the topology design, the studies on topology are typically not material-specific, and there are few limited to a material (3) [80–82]. Material specific design of the topology should give different results with, for instance, *timber* and *steel* because of their different characteristics. *Discrete timber* gridshells are for instance often designed with much longer members and thus coarser grids than *steel* gridshells. More research is therefore suggested on gridshell topologies, particularly linked to a material.

### 4.6. Implications from the Perspective of Architecture

As mentioned in the Introduction, it has been said that the distinction between engineer and architect can be almost artificial when it comes to the design of shell structures; however, as this study shows, the research mapped here is generally driven by structural engineers. Regarding recent examples of built gridshells that push the field forward, like

the Swatch Omega Headquarters by Shigeru Ban [2] (p. 244), and Canary Wharf Crossrail Station by Foster and Partners [2] (p. 219), the architectural idea and how it is presented and perceived is what initiates the realisation of the gridshell. Architectural research on gridshells (as defined in Section 2.2), however, includes very few examples. The contributions from architecture found in this study are generally connected to the *education* of students and typically use *bending active* gridshells as a case. According to this study, it can be fair to say that the architecture field does not drive the traditional research on gridshells. However, by looking into, for instance, architectural magazines, one would find gridshell structures with high architectural quality, where the architectural concept has driven innovation in structural design. Recent examples of this are found in ArchDaily [83], Domus [84] and Structure [85]

If an architect aims to contribute to the field of gridshells, there could be two main directions. The first is to find the gaps in research, typically where the structural engineers are not contributing today. Some possible topics suitable for architects, like bending active metal gridshells and gridshell node design, are already mentioned in this study. The other direction is to start designing. If an architectural design involves a gridshell structure that can enchant the client, this can be a way to realise ambitious gridshells, building on the knowledge created by structural engineers and hopefully pushing the field further.

## 5. Conclusions

This systematic mapping study has investigated recent research on gridshells intending to get an overview of the field. Several databases were used to collect relevant research on gridshell, and a total of 327 papers were used in the mapping. The articles from the mapping increased from ten publications in 2011 to 58 in 2020, indicating a growing interest in gridshell research. The mapping has shown that 278 of the 327 articles on gridshells come from the field of *structural engineering*. In addition, 131 of the 327 mapped articles have been performed as a *theoretical study* that examined a specific *load* or *behaviour*. This corresponds to the keyword mapping, with buckling and finite element analysis among the most occurring keywords. Contributions from architecture were hard to find, indicating that it is not the architectural field that drives the traditional research on gridshells. However, as discussed, architects are instead contributing to the field through the design of gridshells.

The study has identified several knowledge gaps. The clearest is the lack of review articles on *loads* and *behaviour*, which could serve as comprehensive guides for the design and understanding of gridshells for engineers and architects. This is supported by the large number of independent *theoretical studies* on *loads* or *behavior*. Other possible gaps that have been identified are research on *bending active* metal gridshells, material-specific studies on *topology*, research on *nodes* in general and *nodes* for *bending active* gridshells in particular. Seeing these results from the perspective of architecture, all of the mentioned knowledge gaps are research topics where an architect could contribute. For an architect with interest in gridshells, the research future is open. In future work, it would be useful to go deeper into the articles from systematic mapping by extending it to a systematic review (see Petersen et al. [12]), and thus get insight into the content of the articles.

**Author Contributions:** Conceptualization and methodology, S.H.D., B.M. and A.R.; investigation, S.H.D.; writing—original draft preparation, S.H.D.; writing—review and editing, S.H.D. and B.M.; visualization, S.H.D.; supervision, B.M. and A.R. All authors have read and agreed to the published version of the manuscript.

**Funding:** The APC was funded by the Norwegian University of Science and Technology Publishing Fund.

**Institutional Review Board Statement:** Not applicable.

**Informed Consent Statement:** Not applicable.

**Data Availability Statement:** The complete mapping and bibliography used in this study are openly available in [Zenodo] at [https://www.doi.org/10.5281/zenodo.5754722] (accessed on 3 December 2021).

**Acknowledgments:** We thank the two anonymous reviewers whose comments/suggestions helped improve and clarify this manuscript.

**Conflicts of Interest:** The authors declare no conflict of interest.

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
