# Peer review of "Gridshells in Recent Research—A Systematic Mapping Study"

_applsci, doi:10.3390/app112411731_

Round 1

Reviewer 1 Report

In this paper the authors develop a mapping study to investigate all the research carried out about gridshells in the scientific literature since 2011 onwards. The journal articles have been sorted out depending on the scientific fields, authors involved, research type and method, as well as gridshell type and material.

The manuscript is interesting, fits well with the aim of the “Applied Sciences” Journal in general, and especially with the scope of the Special Issue, and can be accepted for publication after the described revisions.

(1) I’d suggest to add a curve related to the cumulative number of papers published since 2011 onwards in Figure 4. In this way, the reader can have a clearer representation of the total number of papers published so far for each year.

(2) Lines 189-190: “Furthermore, the tool/method contributions are also mainly performed as project presentations”. The correct statement should be “Furthermore, the process contributions are also mainly performed as project presentations”.

(3) Lines 198-199: “The most common research motivation was behaviour with 127 articles […]”. However, Figure 10 shows 126 articles for behaviour. 127 should be the correct value, also looking at Figure 8.

(4) Lines 251-252: “Regarding gridshell parts, studies on timber gridshells focus on either the node (18), all parts (13) or the global shape (10)”. The correct statement should be: “Regarding gridshell parts, studies on timber gridshells focus on either the all parts (19) or nodes (4)”.

(5) Lines 446-447: in the Acknowledgments the second and third author are mentioned. This is not necessary, since they share the co-authorship of the work. Therefore, only include relevant people not included into the author list in the Acknowledgements.

According to what said above, the reviewer’s opinion is that the manuscript can be accepted for publication after the described revisions.

Author Response

Thank you for the review. In addition to following up the comments from the editor and the other reviewer, slight changes was made on the "Funding:, Data Availability Statement: & Acknowledgments:"

(1) Thank you for the suggestion. The figure has been updated to include a cumulative number

(2-5) Thank you for picking up these mistakes. The text has been revised accordingly.

Reviewer 2 Report

This work collected, selected, and categorized research articles on gridshells from year 2011 to 2021.  Unlike other publications of review articles in the area of structural engineering, the authors did their work in a systematic way following systematic mapping studies in software engineering of Petersen at al. (2008).  Thus, the paper provides an excellent overview of the recent research on gridshells. Furthermore, the study identified several knowledge gaps that may be fulfilled in future researches.

In the conclusion section, the authors conclude that "the lack of review articles on loads and behaviour, which could serve as comprehensive guides for the design and understanding of gridshells for engineers and architects. ".  This statement is based on from the mapping of published journal articles during 2011-2021.  Actually, there  exists  at least an unpublished research that addressed comprehensive guides for the design and understanding of gridshells, that is, the doctoral dissertation of S.R. Malek (2012) entitled "The Effect of Geometry and Topology on the Mechanics of Grid Shells" https://www.researchgate.net/publication/279817856_The_effect_of_geometry_and_topology_on_the_mechanics_of_grid_shells

Author Response

Thank you for the review. The authors were aware of this research but did not consider including it. It fits very well as an example of existing reviews that was not found in the study (since published as a dissertation)  It has now been added in section 4.4 together with a similar report from the IASS working group.

In addition, comments from the editor and the other reviewer was followed and, slight changes was made on the "Funding:, Data Availability Statement: & Acknowledgments:"